# Carotid Artery Calcifications on Panoramic Radiographs

**DOI:** 10.3390/ijerph192114056

**Published:** 2022-10-28

**Authors:** Joanna Janiszewska-Olszowska, Anna Jakubowska, Ewa Gieruszczak, Kacper Jakubowski, Piotr Wawrzyniak, Katarzyna Grocholewicz

**Affiliations:** 1Department of Interdisciplinary Dentistry, Pomeranian Medical University in Szczecin, 70-111 Szczecin, Poland; 2Independent Researcher, 65-735 Zielona Góra, Poland; 3Independent Researcher, 1230 Vienna, Austria; 4Independent Researcher, 70-515 Szczecin, Poland

**Keywords:** carotid artery, carotid calcifications, carotid bifurcation, panoramic radiograph, dental status

## Abstract

Objective: The objective of this study was to evaluate the prevalence of carotid artery calcifications (CACs) on panoramic radiographs of Polish dental patients. Methods: Panoramic radiographs made between 2005 and 2012 in 4000 patients born between 1922 and 1958 were accessed from the server of the Department of Radiology and analyzed for the presence or absence of CACs by a group of trained dentists. Results: The anatomical area covered by the analysis was visible in 2189 images (54.73%). Calcifications in the carotid arteries were found in 468 (21.68%) patients, including 327 (14.94%) unilaterally and 141 (6.44%) bilaterally. CACs were found in 284 (60.68%) women and 184 (39.32%) men. Conclusions: The prevalence rate of CACs detected on panoramic radiographs in patients aged 54 and older was 21.68%, which makes it an important clinical problem.

## 1. Introduction

Carotid artery calcifications (CACs) on panoramic radiographs (PRs) were first described by Friedlander and Lande [1]. They suggested that panoramic radiographs are a simple and useful aid in indicating patients at high risk of cerebrovascular accident (CVA) due to the visibility of carotid calcifications.

CACs are usually localized in carotid bifurcation. Verticolinear, nodular, or heterogenous nodular radiopaque masses can be identified on panoramic radiographs within the neck soft tissues adjacent to the cervical vertebrae C3–C4 (intervertebral junction) and about 1.5–2.5 cm inferior–posterior to the mandibular angle [2].

Almost 80% of all strokes are ischemic, caused by atherosclerotic disease in the ramification region [3], making the identification of carotid artery calcifications (CACs) of vital importance. Early detection, diagnosis, and relevant medical treatment of those patients who are at risk of cerebrovascular accident (CVA) may reduce the prevalence of strokes, mortality, and morbidity due to CVA [4]. In case of detecting CACs, further diagnostic procedures should be introduced to confirm carotid stenosis (color Doppler ultrasound examination is generally regarded as the gold standard) and enable immediate treatment, if needed.

Dentists should differentiate CACs from anatomical structures such as epiglottis, hyoid bone, calcified stylohyoid ligament, or triticeous and thyroid cartilages, as well as from pathological conditions such as phleboliths, sialoliths, tonsilloliths, and calcified lymph nodes [4,5].

Numerous studies described the use and utility of routine dental panoramic radiographs and the prevalence of CACs on panoramic radiographs in different populations [2,6,7,8,9]. The relevance of PR as a screening tool for patients at risk of CVD has been supported by several observations [10,11,12,13],

The purpose of the present study was to evaluate the prevalence of radiopacities suggestive of carotid artery calcifications on routine dental panoramic radiographs in Polish dental patients.

## 2. Materials and Methods

Four thousand panoramic radiographs of patients born from 1922 to 1958 (2266 women and 1734 men), taken between 2005 and 2012, were obtained from the server of the Department of Radiology after obtaining permission. The inclusion criteria comprised radiographs, which had cervical spine well-visible. CACs were identified within the area of interest that extended 2.5 cm inferior and 2.5 cm posterior of the mandibular angle according to Friedlander and Cohen [14] as presented in Figure 1.

The result was noted as “presence” or “absence“ (binary). Association with sex was assessed using independence test with Yates correction. The Mann–Whitney test was used to analyze age dependence. Values of *p* < 0.05 were considered statistically significant.

## 3. Results

Out of 4000 panoramic radiographs of patients aged 45–87 (mean age: 62 years), 2189 (54.73%, including 1398 women and 798 men) could be included in the present study. The distribution of carotid artery calcifications in the study group has been presented in Table 1.

Calcifications in carotid arteries were found in 468 (21.68%) patients, including 284 women (12.97%) and 184 men (8.4%).

Unilateral calcifications were found in 327 (14.94%) subjects, including 203 (9.27%) women and 124 (5.66%) men, whereas bilateral calcifications were found in 141 (6.44%), including 81 (3.7%) women and 60 (2.74%) men. The average age of women with unilateral calcifications was 63 years, whereas for men it was 64 years. The mean age of women with bilateral calcifications was 61 years, whereas for men it was 65 years. The differences between the sexes proved to be statistically significant for the unilateral CACs (*p* = 0.04), but insignificant for bilateral CACs (*p* = 0.82). The prevalence of unilateral calcification was age-dependent for both sex groups (men *p* = 0.02 and women p = 0.035), but the prevalence of bilateral calcification increased with age for men (*p* = 0.04) but not for women (*p* = 0.42).

## 4. Discussion

Stroke is the second leading cause of death in middle- and high-income countries all around the world [15]. The most important risk factor in stroke is arterial stenosis due to carotid atheroma—an atherosclerotic disease process along the walls of the common carotid artery close to bifurcation. Embolus formed by detached pieces of atheroma may occlude intracerebral artery causing stroke [16].

Considering the general use of panoramic radiographs in most dental practices nowadays, as well as their ease of use and low cost compared with other methods of imaging, PRs can be recognized as an efficient diagnostic tool in CAC observation. Dentists can refer a group of patients at high risk for CVA totally different from the group normally referred by cardiologists or internal medicine doctors. They have the ability to identify thousands of patients at risk of stroke with no additional cost to the health system. Apart from making the CAC patients aware of the risk of stroke, an immediate further and precise type of examination should be recommended, e.g., Doppler ultrasound to confirm or exclude this risk.

While analyzing panoramic radiographs toward the presence of CACs, it is crucial to make careful differential diagnosis. Many cervical radiopacities both physiological (calcified triticeous cartilage, superior corner of calcified thyroid cartilage, greater corner of hyoid bone, styloid process, stylohyoid ligament, stylomandibular ligament, epiglottis) and pathological (calcified lymph node, phleboliths, submandibular salivary gland sialoliths, tonsilloliths) can be easily confused with CACs. Proper differential diagnosis requires knowledge and experience; thus, it seems vital to include the assessment of CACs on panoramic radiographs in the undergraduate dental curriculum.

The present study, with its material of 2189 panoramic radiographs, is one of the largest in Europe. Moreover, it is the first investigation concerning CACs in Poland.

The findings in the literature concerning CACs have been presented in Table 2. The studies found comprised from 83 to 8838 panoramic radiographs. Thus, the present study including 2189 radiographs is one of the largest. Four studies only included European patients (Greek, German, Italian, and Portuguese).

The prevalence of CACs in the present study is similar to that in African women older than 45 [27] and Arabian postmenopausal women [24]. A higher prevalence is reported in studies including older subjects and in patients with systemic diseases including diabetes, hypertension, cancer, or renal diseases [32,35]. In women older than 45, CACs were significantly more prevalent in patients with diabetes and dyslipidemia [27]. A higher prevalence of CACs was found in smokers [21]. Lower prevalence of CACs was found in studies on younger subjects (Table 1). This confirms dependence of the prevalence of CACs on age, sex, and disease.

A possible limitation of the present study is the fact that the authors did not have access to medical records other than panoramic radiographs; thus, no risk factors could be analyzed. However, the authors are aware of the fact that the study group could include patients with systemic diseases and other risk factors.

A correlation has been found between pulp stones [35] and CACs; however, contrary findings were reported in hemodialysis patients and renal transplant recipients [36].

CACs were more prevalent in patients with periodontal disease [2,20,37].

In the present study, significantly more unilateral CACs were similarly found in women as reported by Bayer et al. [18]. The results by Griniatsos et al. [17] as well as by Tamura et al. [7], Beckstrom et al. [2], Tanaka et al. [9], Uthman and Al-Saffar [13], Gonçalves et al. [22], Abreu et al. [19], Santos et al. [30], Maia et al. [33], and Ghassemzadeh et al. [34] confirm a higher prevalence of CACs in women. However, Nasseh and Aoun [28] reported no significant difference between the sexes.

The incomplete dependence of age is contrary to the findings by Ertan and Sisman [4], who found statistically significant differences in CAC rates between age groups. However, they analyzed panoramic radiographs of patients older than 40, whereas at the present study, patients younger than 50 were not included. It is thus possible that age dependence is evident in groups of a wide age range (where younger subjects are included) and diminishes with a narrowing age range. This is confirmed in the studies by de Brito et al. [21], Goncalves et al. [22], Kamak et al. [20], Aghazadehsanai et al. [27], as well as Ribeiro et al. [29].

Side predominance was not analyzed in the present study. The results of other investigations provide contradictory results: left side more prevalent [7], right side predominant [4,6,30,34], and right–left no statistical difference [5,13].

Compared with color Doppler ultrasonography (CDUS), the results of a recent meta-analysis revealed excellent specificity and good sensitivity of panoramic radiographs in the detection of carotid artery calcifications [38]. Ertas and Sisman [4] reported 20.1% false-negative and 18.8% false-positive results; thus, panoramic radiographs had about 80% accuracy, sensitivity, and specificity in the detection of CACs compared with CDUS scans. This confirms that panoramic radiographs have limitations such as practitioners’ inability to quantify the amount of stenosis of a vessel with a recognized radiopacity and the difficulty of accurately diagnosing these lesions as true carotid plaques without advanced training [2]. Constantine et al. [31] have found a carotid stenosis equal to or higher than 50% in every seventh patient with CACs on panoramic radiographs. It is evident that panoramic radiographs cannot be used to make a definitive diagnosis. They may only be used as a supportive screening tool during regular dental treatment.

## 5. Conclusions

Carotid artery calcifications are present in more than 20% Polish dental patients older than 45 years.

Based on the potential of panoramic radiographs to screen CACs, the assessment of carotid calcification should be part of routine analysis of panoramic radiographs by general dentists.

## Figures and Tables

**Figure 1 ijerph-19-14056-f001:**
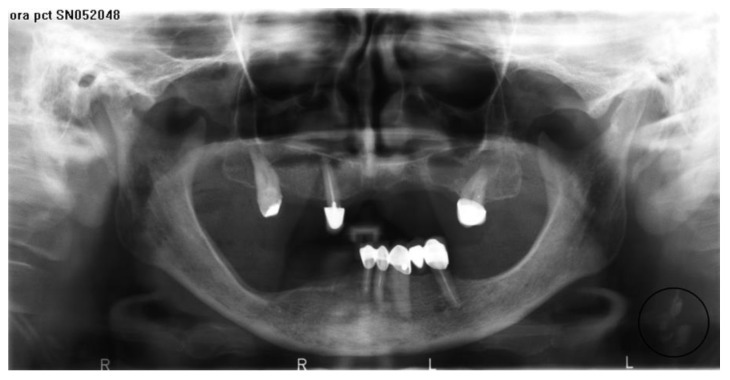
Unilateral carotid artery calcifications (in circle on the left).

**Table 1 ijerph-19-14056-t001:** Distribution of carotid artery calcifications in study group.

CACs Subjects	Unilateral			Bilateral		
	N	%	Mean Age	N	%	Mean Age
Women	203	9.27	63	81	3.7	61
Men	124	5.66	64	60	2.74	65

**Table 2 ijerph-19-14056-t002:** Studies on prevalence of carotid artery calcifications (CACs) on panoramic radiographs.

Author, Year	Study Group Size (Nationality)	Mean Age (Range), Sex	Prevalence of CACs	Unilateral vs. Bilateral
Ravon et al., 2003 [6]	83 (USA)	Men: 69.3 ± 6.7 Women: 68.3 ± 8.3	51 (61.4%)	Unilateral 26 (31.3%); Bilateral: 6 (7.2%)
Tamura et al., 2005 [7]	2568 (Japanese)	63.2 (50–70) Men: 61.9; Women: 62.3	106 (4.13%); Men: 26 (2.13%) Women: 80 (5.94%)	Bilateral 12.3% (7 men, 6 women)
Friedlander and Golub., 2006 [8]	2279 (USA)	Men: 2165, mean age 61.5 (50–86); Women: 114 mean age 63.7 (50–81)	94 Men: 91 Women: 3	Unilateral 39; Bilateral 55
Tanaka et al., 2006 [9]	659 (Japanese)	Older than 80	33 Men: 8 Women: 25	No data
Beckstrom et al., 2007 [2]	201 (USA)	Patients with head or neck cancer 138 men	47 Men: 31 Women: 16	Unilateral 23 (11.4%), 17 men and 6 women; Bilateral 24 (11.9%), 14 men, 10 women
Uthman and Al-Saffar., 2008 [13]	157 (Iraq)	40–80 with chronic illnesses: hypertension, coronary heart disease, diabetes mellitus, hyperlipidemia	61 (38.8%) Men: 32 (40.5%) Women: 29 37.2%	Unilateral: 11 men (13.9%), 10 women (12.8%); Bilateral: 21 men (26.6%), 19 women (24.4%)
Pornprasertsuk-Damrongsri et al. 2009 [16]	85 (Thai)	59.8 (33–75) with metabolic syndrome	19 (22.4%)	Unilateral 11 (12.9%); Bilateral 8 (9.4%)
Griniatsos et al., 2009 [17]	40 (Greek)	Men: 31, mean age 69 (64.5–75); Women: 9, mean age 71 (60–73) with proven carotid artery atherosclerotic occlusive disease	28 (%)	Unilateral 10; Bilateral 18
Bayer et al., 2010 [18]	2557 (German)	64.6 (36–88) Men: 41%; Women: 59%	4.8% Men: 35.2% Women: 64.8%	125 (4.8%) women 81 (64.8%) 44 men (35.2%)
Abreu et al., 2015 [19]	723 (Brazilian)	Minimum 40, mean age with CACs 54.9%	21 (2.9%) Men: 4 (19%) Women: 17 (81%)	Unilateral 13 (61.9%); Bilateral 8 (38.1%)
Kamak et al., 2015 [20]	1146 (Turkish)	53.84 ± 9.25 Men: 577 (50.3%); Women: 569 (49.7%)	156 (13.6%) including 40 (3.5%) right side and 52 (4.5%) left side	Unilateral 92 (7%); Bilateral 64 (5.6%)
de Brito et al., 2016 [21]	505 (Brazilian)	50.1 (30–80)	7.2% Men: 6.53% Women: 8.82%	No data
Gonçalves et al., 2016 [22]	8338 (no data)	34 (4–94)	576 (6.90%) Men: 216 (6.57%) Women: 363 (7.19%)	Right: 180 (2.15%|, left: 182: (2.18%); Bilateral: 127 (2.6%)
Patil et al., 2016 [23]	240 (Saudi Arabian), 120 with renal stoned vs. 120 controls	Renal stone patients: mean age 40; controls: mean age 41	25 (20.8%) in renal stone patients and 16 (12.3%) in control group	No data
Patil, 2017 [24]	1214 (Saudi Arabian)	Older than 50 62.47 ± 5.27 (postmenopausal women)	278 (22.9%)	Right 102 (36.7%), left 126 (45.3%); Bilateral: 50 (18%)
Friedlander et al., 2017 [25]	531 (USA)	Men with gout	163 (31%)	No data
Markman et al., 2017 [26]	180 (Brazilian)	59.4 (20–85) with head and neck cancer	Both radiographs 57 (31.67%)	No data
Aghazadehsanai et al., 2017 [27]	171 (African American)	Women older than 46	41 (24%)	No data
Nasseh and Aoun, 2018 [28]	500 (Lebanese)	Age range 18–88	34 (6.8%)	Unilateral: 12 (2.4%); Bilateral: 22 (4.4%)
Ribeiro et al., 2018 [29]	2375 (Portuguese)	38 (3–90)	18 (5.1%)	No data
Santos et al., 2018 [30]	2500 (Brazilian)	54 (18–89)	96 (4%)	Unilateral: 59 (2.36%); Bilateral: 37 (1.48%)
Constantine et al., 2019 [31]	5780 (Australian)	Over 18 years old	623 (10.8%)	No data
Agacayak et al., 2020 [32]	284 (Turkish with systemic diseases (hypertension, diabetes, coronary artery disease, atherosclerosis, hyperlipidemia)	66 (60–92) Men: 204; Women: 240	39 (8.8%)	No data
Maia et al., 2021 [33]	1176 (Brazilian)	67, at least 60	147 (12.5%) Men: 64 (17.7%) Women: 83 (11.3%)	No data
Ghassemzadeh et al., 2021 [34]	2307 (Italian referred for dental treatment, treatment of trauma, for bisphosphonates therapy, transplant or cardiac intervention)	47	167 (31.57)	No data
Present study	2189 (Polish)	62 (45–86) Women: 2266; Men: 1734	468 (21.68%), Men: 184 (8.4%) Women: 284 (12.97%)	Unilateral: 327 (14.94%); Bilateral: 141 (6.44%)

## Data Availability

Raw data can be obtained from the corresponding author on a reasonable request.

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
