# Peer review of "Carotid Artery Calcifications on Panoramic Radiographs"

_ijerph, 2022, doi:10.3390/ijerph192114056_

Round 1
Reviewer 1 Report
This study is interesting and significant especially in terms of the number of people used for assessment. A few things however must be corrected and cleared.
Please clearly mention that the outcome is binary (presence, absence) scale.
Sometimes "," was placed in decimal point.
Authors didn't clearly mention how 2189 people were included out of 4000. What were the inclusion criteria?
Authors mentioned that they grouped people by age. How they were grouped, how many groups, how many people per group?
In materials and methods, the term "correlation" should be changed to "association".
In Table 1, the denominators for calculating the rate were not provided. How many people were female or male of 2189?
In conclusion, authors concluded that about 20% of people had CAC. Please clearly mention the inclusion criteria the authors considered for assessment.
Second conclusion I think is out of scope of the study. Including the assessment in undergraduate dental program sounds too much technical for the program. Authors also mentioned that accurate assessment needs advanced training.
Author Response
Dear Reviewer
This study is interesting and significant especially in terms of the number of people used for assessment. A few things however must be corrected and cleared.
Thank you for giving us the opportunity to submit a revised draft of our manuscript to the International Journal of Environmental Research and Public Health. We appreciate the time and effort you have dedicated to provide your valuable feedback on our manuscript. We are grateful for your insightful comments on our paper. We were able to incorporate changes to reflect most of the suggestions provided. We have highlighted the changes in the manuscript.
Here is a point-by-point response to the comments and concerns.
Comment 1: Please clearly mention that the outcome is binary (presence, absence) scale.
Response: Thank You for pointing this out. According to Your suggestion we have added the explanation to the manuscript.
Comment 2: Sometimes "," was placed in decimal point.
Response: Thank You for Your comment, adjustments have been made as recommended.
Comment 3: Authors didn't clearly mention how 2189 people were included out of 4000. What were the inclusion criteria?
Response: The inclusion criteria comprised radiographs, which had cervical spine well visible. This information is placed in the material and methods section.
Comment 4: Authors mentioned that they grouped people by age. How they were grouped, how many groups, how many people per group?
Response: Thank You for Your comment. No age groups were formed, suitable correction has been made in the manuscript.
Comment 5: In materials and methods, the term "correlation" should be changed to "association".
Response: This change has been made as kindly recommended.
Comment 6: In Table 1, the denominators for calculating the rate were not provided. How many people were female or male of 2189?
Response: Thank You for Your comment, adjustments have been made as recommended.
Comment 7: In conclusion, authors concluded that about 20% of people had CAC. Please clearly mention the inclusion criteria the authors considered for assessment.
Response: The spelling error has been corrected, thank You for Your comment.
Comment 8: Second conclusion I think is out of scope of the study. Including the assessment in undergraduate dental program sounds too much technical for the program. Authors also mentioned that accurate assessment needs advanced training.
Response: Thank You for pointing this out. According to your suggestion we have changed the second conclusion.
Best regards,
Authors
Reviewer 2 Report
This manuscript submitted by Joanna Janiszewska-Olszowska et al., addresses the potential of routine dental panoramic radiographs (PR) to identify CAC. The paper report on a retrospective observational study that enrolled a cohort of Polish dental patient aged 45-87 years of age.
The importance of these type of studies relies on the general use of PR’s in dental practice and on recent findings associating carotid calcifications in PR’s with risk of stroke or ischemic heart diseases.
Concerning the present study, the most relevant feature is to report on a Polish cohort for the first time, thus providing the scientific community with additional regional data. On this basis the paper should be published.
However, the following issues should be considered before acceptance:
1 - The relevance of PR as a screening tool for patients at risk of CVD would benefit of a more updated overview. A quick search on recently publish data yield a few interesting papers which may deserve an overview by the authors (e.g., https://doi.org/10.1007/s00784-018-2533-8; https://doi.org/10.1259/dmfr.20160147; https://doi.org/10.1111/idj.12618).
2 – As mentioned in Methods and Results age groups were used. Authors should describe which age intervals were considered and how the prevalence of age-dependent CAC were distributed in the study cohort according to the age groups. A figure or a table with these results would significantly improve the paper relevance.
3 – Revise ln 99. Table 2 organization and data included would deserve a concise description in the text.
4 – Conclusions are rather poor. Authors should be able to summarize the relevant arguments referred in discussion to support the inclusion of CAC identification in undergraduate dental curricula: demonstrate the potential of PR to screen CAC in several populations across the world; the importance of introducing a new tool, which is beneficial in terms of health costs, to referral of patients for outpatient’s clinic of CVD and/or further medical examinations. From my point of view these two issues should be emphasized.
Other remarks:
1 - Data in Table 2 should be harmonized, e.g., significant digits, “no data” whenever there is no data available, etc..
2 – When comparing data obtained in the present study to published data the pathology or health condition have to be mentioned, e.g. ln 104-105.
3 – Check acronyms definition throughout the manuscript (e.g., ln 140); check statistical terminology (e.g., insignificant ln 71); check decimal separator (ln 73).
4 – Consider to include a typical radiograph to illustrate the detection of CAC (and other features that may require the implementation of training in undergraduate dental courses).
Author Response
Dear Reviewer
This manuscript submitted by Joanna Janiszewska-Olszowska et al., addresses the potential of routine dental panoramic radiographs (PR) to identify CAC. The paper report on a retrospective observational study that enrolled a cohort of Polish dental patient aged 45-87 years of age.
Thank you for giving us the opportunity to submit a revised draft of our manuscript to the International Journal of Environmental Research and Public Health. We appreciate the time and effort you have dedicated to provide your valuable feedback on our manuscript. We are grateful for your insightful comments on our paper. We were able to incorporate changes to reflect most of the suggestions provided. We have highlighted the changes in the manuscript.
Here is a point-by-point response to the comments and concerns.
The importance of these type of studies relies on the general use of PR’s in dental practice and on recent findings associating carotid calcifications in PR’s with risk of stroke or ischemic heart diseases.
Thank You for this positive comment.
Concerning the present study, the most relevant feature is to report on a Polish cohort for the first time, thus providing the scientific community with additional regional data. On this basis the paper should be published.
Thank You for Your positive opinion on our study.
However, the following issues should be considered before acceptance:
Comment 1: The relevance of PR as a screening tool for patients at risk of CVD would benefit of a more updated overview. A quick search on recently publish data yield a few interesting papers which may deserve an overview by the authors (e.g., https://doi.org/10.1007/s00784-018-2533-8; https://doi.org/10.1259/dmfr.20160147; https://doi.org/10.1111/idj.12618).
Response: Thank You for pointing this out. According to your suggestion we have added the articles confirming the importance of PRs to the manuscript
Comment 2: As mentioned in Methods and Results age groups were used. Authors should describe which age intervals were considered and how the prevalence of age-dependent CAC were distributed in the study cohort according to the age groups. A figure or a table with these results would significantly improve the paper relevance.
Response: Thank You for your comment. No age groups were formed, suitable correction has been made in the manuscript.
Comment 3: Revise ln 99. Table 2 organization and data included would deserve a concise description in the text.
Response: Thank you for this important comment. A concise description to Table 2 has been added, as kindly proposed. The discussion of the results of the present investigation compared to previous studies is placed below the table.
Comment 4: Conclusions are rather poor. Authors should be able to summarize the relevant arguments referred in discussion to support the inclusion of CAC identification in undergraduate dental curricula: demonstrate the potential of PR to screen CAC in several populations across the world; the importance of introducing a new tool, which is beneficial in terms of health costs, to referral of patients for outpatient’s clinic of CVD and/or further medical examinations. From my point of view these two issues should be emphasized.
Response: Thank You for Your comment. Addressing the comment by the other reviewer, the second conclusion has been revised as follows:
“Basing on the potential of panoramic radiographs to screen CACs, the assessment of carotid calcification should be part if routine analysis of panoramic radiographs by general dentists.”
Other remarks:
1 - Data in Table 2 should be harmonized, e.g., significant digits, “no data” whenever there is no data available, etc..
Response: Thank You for Your comment, adjustments have been made as recommended.
2 – When comparing data obtained in the present study to published data the pathology or health condition have to be mentioned, e.g. ln 104-105.
Response: Thank You for the comment. The authors have placed information on the health condition of patients included in previous papers in the following sentences:
“The prevalence of CACs in the present study is similar as in African women older than 45 [27] as well as in Arabian postmenopausal women [24]. A higher prevalence is reported in studies including older subjects and in patients with systemic diseases including: diabetes, hypertension, cancer or renal diseases [32, 35]. In women older than 45, CACs were significantly more prevalent in patients with diabetes and dyslipidemia [27]. A higher prevalence of CACs was found in smokers [21].”
3 – Check acronyms definition throughout the manuscript (e.g., ln 140); check statistical terminology (e.g., insignificant ln 71); check decimal separator (ln 73).
Response: Thank You for Your comment. Suitable correction has been made in the manuscript. The acronym CDUS (for color Doppler ultrasonography) was used in the study cited. The word “insignificant” was used as a synonim for “non-significant”. Decimal separator has been corrected for point, as recommended.
4 – Consider to include a typical radiograph to illustrate the detection of CAC (and other features that may require the implementation of training in undergraduate dental courses).
Response: Thank You for Your suggestion. A figure has been added, as recommended.
Best regards,
Authors
Round 2
Reviewer 2 Report
Dear Authors
Thank you for considering my comments and change the manuscript accordingly.
From my point of view, your paper can be published as it is.